Patient exposure in the basic science classroom enhances differential diagnosis formation and clinical decision-making

Peacock Justin G. 1 justin.g.peacock@gmail.com
Grande Joseph P. 2
1 Department of Graduate Medical Education, San Antonio Uniformed Services Health Education Consortium , San Antonio, TX , USA
2 Department of Laboratory Medicine and Pathology, Mayo Clinic College of Medicine , Rochester, MN , USA
Perry George
Electronic publication date: 2015 Feb 26
Publication date: 2015
Volume: 3
Electronic Location ID: e809
Received 2014 Dec 20; Accepted 2015 Feb 10
Copyright: © 2015 Peacock and Grande
Copyright year: 2015
Copyright holder: Peacock and Grande
License: This is an open access article distributed under the terms of the Creative Commons Attribution License, which permits unrestricted use, distribution, reproduction and adaptation in any medium and for any purpose provided that it is properly attributed. For attribution, the original author(s), title, publication source (PeerJ) and either DOI or URL of the article must be cited.
License URL: https://creativecommons.org/licenses/by/4.0/

Keywords: Patient exposure, Pre-clinical, Undergraduate medical education, Differential diagnosis, Clinical decision-making, Pathology, Basic science, Clinical skills, History-taking, Empathy

Funding: The authors declare that there was no funding for this work.

==============================
Purpose. The authors proposed that introducing real patients into a pathology classroom early in medical education would help integrate fundamental principles and disease pathology with clinical presentation and medical history.

Methods. Three patients with different pathologies described their history and presentation without revealing their diagnosis. Students were required to submit a differential diagnosis in writing, and then were able to ask questions to arrive at the correct diagnosis. Students were surveyed on the efficacy of patient-based learning.

Results. Average student scores on the differential diagnosis assignments significantly improved 32% during the course. From the survey, 72% of students felt that patient encounters should be included in the pathology course next year. Seventy-four percent felt that the differential diagnosis assignments helped them develop clinical decision-making skills. Seventy-three percent felt that the experience helped them know what questions to ask patients. Eighty-six percent felt that they obtained a better understanding of patients’ social and emotional challenges.

Discussion. Having students work through the process of differential diagnosis formulation when encountering a real patient and their clinical presentation improved clinical decision-making skills and integrated fundamental concepts with disease pathology during a basic science pathology course.

Introduction

A major challenge during medical school is making the leap from the basic science years to patient encounters in the clinical setting. One difficulty involves forming differential diagnoses for a patient’s problem. Forming a differential diagnosis is critical to ordering proper clinical testing and appropriately managing clinical disease. Unfortunately, most differential diagnosis education comes during the third-year clerkships when the medical student is thrown into unfamiliar clinical situations and responsibilities. Few attempts have been reported in the literature aimed at teaching differential diagnosis and clinical decision-making in the early “basic science” years of medical education (Fulop, 1985; Duque, Gold & Bergman, 2003; Gesundheit et al., 2009; Jacobson et al., 2010; Gunning & Fors, 2012).

Some approaches in “early clinical exposure” have focused on the use of clinical cases and standardized or virtual standardized patients to train medical students in clinical reasoning (Gesundheit et al., 2009; Jacobson et al., 2010; Gunning & Fors, 2012). The use of clinical cases and standardized patients has been well studied and documented in the medical education literature (Tamblyn et al., 1991a; Tamblyn et al., 1991b; Ainsworth, 1991; Colliver et al., 1998; Williams, 2004). While standardized patients have a long history in medical education, their use also has identified problems, including subjective biases in the standardized patients, inaccuracies, and unrealistic portrayals of patient experiences (Tamblyn et al., 1991a; Tamblyn et al., 1991b; Williams, 2004).

At the Mayo Medical School, we have been teaching differential diagnosis in the context of the first-year Pathology course for a number of years. Students are taught how to form differential diagnoses and to begin the initial steps of clinical decision-making (Martin et al., 2014). In the past, differential diagnosis was taught in the context of short patient cases that illustrated common human pathologies. We wanted to try a pilot study to bring actual patients with true pathologies into the basic science classroom, early in undergraduate medical education, to see if students could delineate a differential diagnosis for their pathologies. We wanted to help students integrate fundamental principles of the basic sciences with a patient’s clinical presentation, disease pathology, and course of disease. We also wanted to increase students’ awareness of the important aspects of clinical histories and appropriate diagnostic testing or questioning in arriving at an accurate diagnosis. We hypothesized that actual patients with clinical pathologies would provide even more important learning with regard to clinical decision-making for the students.

Materials and Methods

This study was submitted to the Mayo Clinic Institutional Review Board for approval. It was determined in verbal communications not to require IRB approval because it was conducted in the context of required assignments in an educational course.

In this pilot study, 47 first-year pathology students were introduced to three different volunteer patients with actual disease pathologies to present their clinical histories. The patients had previously expressed an interest in sharing their clinical history with medical students to faculty and staff at the clinic. The three patients came into the classroom on three different occasions during the course and spent approximately 30 min discussing the history and clinical scenario leading up to their diagnosis, without revealing their diagnosis. Immediately after the patient finished their history, students then completed an online assignment, which asked for their top three diagnoses (beginning with the most likely), an explanation of the aspects of the case that led to their most likely diagnosis, and additional questions or information that would help to confirm their differential (Fig. 1). The goal with the questions was not for the students to obtain the exact, true diagnosis, but for them to use their clinical reasoning skills to narrow down a reasonable differential diagnosis list based on the patient’s history. We wanted the students to justify the rationale for their differential list and to formulate additional questions or tests that they would like to use to narrow down the differential list. Following submission of the online assignment, students were then permitted to ask the patients additional questions to determine the true diagnosis as a class. They were also given time to ask questions regarding the social, behavioral, economic, and other impacts of the disease on the patients’ lives.

Figure 1 Questions and grading rubric for patient encounters.

The patient encounter questions given to the students are seen in black type, while the grading rubric is seen in red type.

As the patient cases were not straightforward, simple, first-year cases, it was stressed to both the students and the teaching assistants that the goal of the assignments was not to obtain the exact, correct diagnosis, but to formulate a reasonable differential diagnosis list based on the patient’s history. We specifically sought out patients with more complicated or multi-system diagnoses to expand the differential diagnoses that students could assemble for a given patient history. Consequently, the cases and the grading rubric allowed for a wider variability in the differential diagnoses compared with simpler medical case presentations. Teaching assistants utilized an established grading rubric and grading methodology used in other differential diagnosis assignments in the course to grade the assignments on an 18-point scale (Fig. 1) (Martin et al., 2014). Student scores on the assignments were compared using ANOVA with Tukey–Kramer post-processing at the 0.05 α level.

At the conclusion of the course, a survey was conducted among the students to determine the impact of patient encounters on their understanding of pathology, differential diagnosis formation, clinical reasoning, and patient empathy. The survey results were tallied on the basis of a five-point Likert scale. For simplicity in the summary table, strong and very strong agreement ratings were grouped into the agreement column and strong and very strong disagreement ratings were grouped into the disagreement column.

Results

We piloted bringing volunteer patients with actual disease into a first-year pathology course to help students develop their differential diagnosis, history-taking, and basic clinical decision-making skills. The students’ previous first-year courses included genetics, anatomy, and histology. The first volunteer patient presented his history of liver cirrhosis secondary to alpha-1 antitrypsin deficiency during the first week of the course. The differential assignments were graded by teaching assistants using the grading rubric shown in Fig. 1 and using a previously described grading methodology (Martin et al., 2014). The first assignment resulted in an average score of 8.8 ± 3.1 out of 18 (Fig. 2). During the pathology course, lectures, assignments, and teaching assistant feedback was devoted to helping students learn how to develop differential diagnoses, including the VITAMIN CDE methodology (Martin et al., 2014), how to ask appropriate questions of patients, and how to make basic diagnostic decisions.

Figure 2 Box plot of student scores for patient encounter assignments.

Box plot with 25% quartiles and median for the grades of students during the first, second, and third patient encounters. One-factor ANOVA with repeated measures indicated a significant difference in the grades between the 3rd patient scores compared with the 1st and 2nd patient scores [F Ratio = 12.1244, P < 0.0001]. ∗P < 0.05, by post hoc Tukey-Kramer HSD. Significant pairings are designated by a bracket connecting the pairings with an * above the bracket.

The second volunteer patient was a pediatric patient who experienced a biliary leak and infection secondary to liver transplantation. The mother and patient presented the patient’s history to the class approximately 3 weeks after the first patient. The second assignment resulted in an average score of 9.7 ± 2.7 (Fig. 2), which was higher than the first assignment, but not significantly higher. Due to scheduling conflicts, the third patient presented his case of heart failure secondary to hypertrophic obstructive cardiomyopathy approximately one week after the second patient. The average score on this third assignment was 11.6 ± 2.7, which was significantly higher than both the first and second assignments, p < 0.0001 and p = 0.0043, respectively (Fig. 2). We found that the students’ grades on the patient encounter differential diagnosis assignments improved significantly by 32% over the course of the block (Fig. 2).

At the end of the pathology block, students were asked to fill out a survey regarding their experiences with the patient encounter differential diagnosis assignments (Table 1). Overall, the students felt that the patient encounter experiences should be a continuing part of the pathology block and other first-year medical school courses (72.3% and 66.0%, respectively). Importantly, students strongly felt that the patient encounter experiences helped them develop clinical decision-making skills, know what questions to ask patients, and understand social and emotional challenges that patients face during disease (74.5%, 74.5%, and 87.2%, respectively). Interestingly, the students still indicated that they did not feel more comfortable facing patient encounters in the clinic after these experiences (27.7%). Students also indicated that they would have liked to have had patients with simpler, more common pathologies than those presented (data not shown).

Table 1 Survey summary for student survey regarding patient encounter experiences.

Survey statistics are listed as percentages of the total class responses (N = 47). Likert scores of strong/very strong agree (4/5) are grouped together in agreement column, Likert scores of strong/very strong disagree (1/2) are grouped together in disagreement column, and the rest (Likert score 3) are in the neutral column.

Survey statement	Agreement	Neutral	Disagreement	
Patient encounters should be incorporated into the Pathology block next year.	72.3	12.8	14.9	
More patient encounters should be included in the first-year courses.	66.0	17.0	17.0	
The DDX assignments associated with the patient encounters help me develop differential diagnosis formation skills.	57.4	17.0	23.4	
The DDX assignments associated with the patient encounters helped me develop clinical decision-making skills.	74.5	10.6	14.9	
I have a better understanding of pathology through the patient encounter experiences.	53.2	21.3	25.5	
The patient encounters helped me to better link a patient’s pathology with the patient’s clinical presentation.	68.1	14.9	14.9	
The patient encounters helped me to better recognize key elements of clinical history and exam in a patient.	66.0	23.4	10.6	
The patient encounters improved my confidence to interact with patients.	27.7	31.9	40.4	
The patient encounters gave me a better sense of what questions would be important to ask patients.	74.5	12.8	12.8	
The patient encounters gave me a better sense of what diagnostic procedures or test to order for patients.	63.8	6.4	29.8	
The patient encounters gave me a better appreciation of the social and emotional challenges that patients go through.	87.2	8.5	4.3	
The patient encounters helped to give you insight into what your clinical experience might be like.	68.1	19.1	10.6	

Discussion

In this study, we have demonstrated that real patient encounters in the basic science classroom coupled with assignments aimed at clinical decision-making may improve clinical skills and help to provide clinical context to the basic sciences they are learning. We have shown that actual, volunteer patients can be brought into a first-year, basic science classroom to serve as patient educators, helping students learn to ask the right questions, formulate differential diagnoses, and understand the nonmedical challenges that patients endure. We believe that this coupling of the basic and clinical sciences in the early years of medical school is important to help students more readily and confidently transition from the classroom to the clinical setting.

Other institutions have provided beginning medical students with early clinical exposure, through the use of clinical case scenarios and simulated patients (Fulop, 1985; Duque, Gold & Bergman, 2003; Gesundheit et al., 2009; Jacobson et al., 2010; Gunning & Fors, 2012). The challenge with these simulated experiences lies in their very nature, namely, they are simulated or artificial experiences (Tamblyn et al., 1991a; Tamblyn et al., 1991b; Williams, 2004). Actual patient encounters provide realistic exposure to clinical scenarios. They can provide clinical context and psychosocial factors that cannot be considered in typical standardized patient scenarios. We did not focus on the students coming up with the exact, correct diagnosis at this early stage, because we felt that the process involved in formulating and justifying a reasonable differential diagnosis list is more important in early medical education. In particular, we wanted students to broaden their differential in the context of the pathology that they were learning about in the course.

Another benefit of real patient exposure in the early medical school years lies in the development of empathy or emotional IQ. While not directly assessed in this study, an overwhelming percentage of students did report a greater appreciation for the social and emotional challenges that patients endure. It is not clear that the same increase in empathy could be obtained from simulated patient experiences (Colliver et al., 1998). Empathy or emotional IQ is an important attribute of professional physicians, particularly, with many studies showing a decline in empathy over the course of medical education (Neumann et al., 2011).

We understand that our study is limited in that it was a pilot study for bringing actual patients into the basic science classroom. We only were able to recruit the three patient volunteers during the first round of recruiting. While we felt that each patient scenario was appropriate for allowing students to create a reasonable differential diagnosis list, we were not able to choose from among a wide variety of patient scenarios. We also encountered some of the challenges in dealing with actual patients rather than simulated patients, in that we were not able to decide when students would encounter patient scenarios due to patient schedules. We also did not have fine control over how much or how little the patients would discuss about their cases. The patients were instructed to give as much pertinent history as they could without revealing the actual diagnoses being assessed. In the future, we hope to have a larger pool of volunteer patients from which to decide the most appropriate patient encounters at regular intervals during the course.

Interestingly, although students felt that the patient encounters increased their clinical decision-making skills and helped them to know what types of questions to ask patients, many students still did not feel an increase in confidence in dealing with patients (Table 1). Understanding that these students are still first-year students, it would be interesting to know what knowledge, skills, and experiences students require to feel confident going into clinical patient scenarios. It would also be interesting to see how the patient educators would rate students with regards to empathy or emotional IQ during these sessions to work on improving this professional characteristic. Unfortunately, in this study, we did not collect a grade breakdown for the different grading rubric criteria from the teaching assistants, only a total grade for the assignment. In the future, we would like to understand what specific areas showed the most improvement during the course, i.e., differential formation, differential justification, or clinical decision-making. Lastly, we would like to develop objective means to determine if the skills taught in this course are being implemented in the clinical setting.

Conclusions

Introducing actual patients into a first-year, basic science pathology classroom helped students to develop differential diagnosis formation, history-taking, and basic clinical decision-making skills at an early stage of undergraduate medical education. Students also reported that they were better able to understand and appreciate the challenges that patients face during the course of their disease.

Disclaimer

The view(s) expressed herein are those of the author(s) and do not reflect the official policy or position of Brooke Army Medical Center, the U.S. Army Medical Department, the U.S. Army Office of the Surgeon General, the Department of the Army, the Department of the Air Force and Department of Defense or the U.S. Government.

Supplemental Information

Table S1 Raw data for Table1

Survey statistics used in Table 1. Statistics are listed as percentages of the total class responses (N = 47). Questions pertinent to this study are included here.

Click here for additional data file.

Figure S12 Raw data for Fig. 2

Excel table containing the individual grades (column 2) for the three patient assignments (column 1).

Click here for additional data file.

Additional Information and Declarations

Competing Interests

Author Contributions

Human Ethics

The authors declare that they have no competing interests.

Justin G. Peacock conceived and designed the experiments, performed the experiments, analyzed the data, contributed reagents/materials/analysis tools, wrote the paper, prepared figures and/or tables, reviewed drafts of the paper, recruited the patient volunteers.

Joseph P. Grande conceived and designed the experiments, performed the experiments, analyzed the data, contributed reagents/materials/analysis tools, reviewed drafts of the paper.

The following information was supplied relating to ethical approvals (i.e., approving body and any reference numbers):

1. Mayo Clinic College of Medicine IRB

2. The IRB determined, in verbal communications, that it did not require IRB approval, because the study was conducted in the context of required assignments in an educational course.

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
