# Peer review of "Patient exposure in the basic science classroom enhances differential diagnosis formation and clinical decision-making"

_PeerJ, doi:10.7717/peerj.809_

## Round 0.1 · original submission · Minor Revisions

Please address the noted minor issues and provide a response

Reviewer 1 ·

Basic reporting

Word choice error par. 3 of results, line 8. "Interestingly, the students still indicated that they did not fill more comfortable facing patient
encounters in the clinic after these experiences (27.7%)." "Fill" should be replaced with "feel"

Experimental design

No comments

Validity of the findings

No comments

Additional comments

The study was clear and easy to understand

·

Basic reporting

It would be interesting to know which of the 3 grading criteria showed improvement over the course of the 3 patients; i.e. was the diagnosis better or did they develop a better understanding of the questions to ask?

Experimental design

No comment

Validity of the findings

No comment

Additional comments

Interesting study. Do you think having the students in small groups would have improved their confidence when they got to the clinic?

Reviewer 3 ·

Basic reporting

The article meets the standards

Experimental design

It is in the area that I have problems. The author make the case that exposing students to real patients in the classroom will better prepare students. In this case was pathology. Discipline based education research indicates that this is the case (Doyle, 2008). So, in principle, the authors intend to test the importance of relevancy in learning. However, one thing that is lacking in the design is an appropriate control. Can the authors use past exam scores and compare them to the current results? It seems that this will not be difficult to do.

Validity of the findings

The statistical tests were appropriate

Additional comments

The author should compare the current results with the scores from previous students who were not exposed to the real patients.

---

## Round 0.2 · accepted · Accept

Thank you for addressing the critique